# FLoRA: Federated Fine-Tuning Large Language Models with Heterogeneous Low-Rank Adaptations

**Ziyao Wang[1], Zheyu Shen[1], Yexiao He[1], Guoheng Sun[1] Hongyi Wang[2,3]**
**Lingjuan Lyu[4], Ang Li[1]**
1. University of Maryland, College Park
2. Rutgers University 3. GenBio.ai 4. Sony Reasearch
`{ziyaow,zyshen,yexiaohe,ghsun,angliece}@umd.edu,`
`hongyi.wang.001@rutgers.edu, lingjuanlvsmile@gmail.com`

## Abstract

The rapid development of Large Language Models (LLMs) has been pivotal in advancing AI, with pre-trained LLMs being adaptable to diverse downstream tasks through fine-tuning. Federated learning (FL) further enhances fine-tuning in a privacy-aware manner by utilizing clients' local data through in-situ computation, eliminating the need for data movement. However, fine-tuning LLMs, given their massive scale of parameters, poses challenges for clients with constrained and heterogeneous resources in FL. Previous methods employed low-rank adaptation (LoRA) for efficient federated fine-tuning but utilized traditional FL aggregation strategies on LoRA adapters. These approaches led to mathematically inaccurate *aggregation noise*, reducing fine-tuning effectiveness and failing to address heterogeneous LoRAs. In this work, we first highlight the mathematical incorrectness of LoRA aggregation in existing federated fine-tuning methods. We introduce a new approach called FLoRA that enables federated fine-tuning on heterogeneous LoRA adapters across clients through a novel stacking-based aggregation method. Our approach is noise-free and seamlessly supports heterogeneous LoRA adapters. Extensive experiments demonstrate FLoRA's superior performance in both homogeneous and heterogeneous settings, surpassing state-of-the-art methods. We envision this work as a milestone for efficient, privacy-preserving, and accurate federated fine-tuning of LLMs. Our code is available at `https://github.com/ATP-1010/FederatedLLM`.

## 1 Introduction

The Large Language Models (LLMs) have shown remarkable performance on various tasks, such as chatbots [1], virtual assistants [4], search engines [11], and healthcare [20; 18]. However, adapting pre-trained LLMs (*e.g.,* Llama 2 [22]) to downstream tasks requires tremendous computation resources to fine-tune all the model parameters. To mitigate this issue, a variety of parameter-efficient fine-tuning (PEFT) methods have been proposed. One of the most widely used PEFT methods is low-rank adaptation (LoRA) [10]. As shown in the top of Figure 1, LoRA adds a parallel branch of trainable adapters $\mathbf{A}$ and $\mathbf{B}$ to compute the model update $\mathbf{\Delta W}$, where the ranks of $\mathbf{A}$ and $\mathbf{B}$ are much smaller than the pre-trained model parameter $\mathbf{W}$. When applying LoRA for fine-tuning, only $\mathbf{A}$ and $\mathbf{B}$ are updated while the entire $\mathbf{W}$ is frozen, thereby significantly reducing the GPU memory consumption.

Fine-tuning LLMs requires ample data for adaptation to specific downstream tasks [13; 7]. Often, this data is dispersed across a multitude of devices, raising privacy concerns. For instance, aggregating medical data from hospitals for centralized LLM fine-tuning poses significant challenges. Consequently, to facilitate fine-tuning without compromising private data, federated learning (FL) becomes essential, enabling LLM fine-tuning across distributed clients while preserving data privacy [16; 27; 33; 24]. In this work, we focus on federated fine-tuning, enabling distributed clients to collaboratively fine-tune LLMs for adaptation to downstream tasks while preserving data privacy.

38th Conference on Neural Information Processing Systems (NeurIPS 2024).

Prior work, FedIT, proposed a federated fine-tuning method [28], integrating LoRA with FedAvg [16]. In each FL round of FedIT, clients fine-tune LoRA modules using their local data and then send the fine-tuned modules to the server. The server averages all the local LoRA modules to obtain a global LoRA. Since only the weights of the LoRA modules are fine-tuned and communicated, FedIT effectively reduces both computation and communication costs.

However, FedIT faces two key issues. First, **the naive averaging of local LoRA modules in FedIT introduces noise to the global model update.** Specifically, FedIT averages local $\mathbf{A}$ and $\mathbf{B}$ independently, which introduces mathematical errors to the global LoRA. In short,

The cause of aggregation noise:

$$\underbrace{\sum \mathbf{A} \times \sum \mathbf{B}}_{\text{FedIT}} \neq \underbrace{\sum \mathbf{A} \times \mathbf{B}}_{\text{mathematically correct}}.$$

We will elaborate on this issue in Section 2 with theoretical analysis. Such an inaccurate aggregation will hinder convergence, leading to higher fine-tuning costs. Second, due to the heterogeneous data distribution [31; 12] and heterogeneous hardware resources, clients need to adapt LoRA ranks [30] according to the system and data heterogeneity. However, **FedIT cannot aggregate local LoRAs with heterogeneous ranks.**

In this work, we present FLoRA, an aggregation-noise-free federated fine-tuning method that supports heterogeneous LoRAs. Specifically, as shown in Figure 2, we propose to **stack** the local LoRA modules $\mathbf{A}_k$ and $\mathbf{B}_k$ separately to construct the global LoRA modules $\mathbf{A}$ and $\mathbf{B}$, where $\mathbf{A}_k$ and $\mathbf{B}_k$ denote the corresponding LoRA modules on the $k$-th client. This stacking method is theoretically proven to be accurate for the aggregation of local LoRA modules (Section 3.1). Additionally, it can naturally accommodate heterogeneous LoRA settings (Section 3.2), since stacking does not require the local LoRA modules to have identical ranks across clients. The noise-free aggregation of FLoRA accelerates convergence, which will in turn improve the overall computation and communication efficiency of federated fine-tuning.

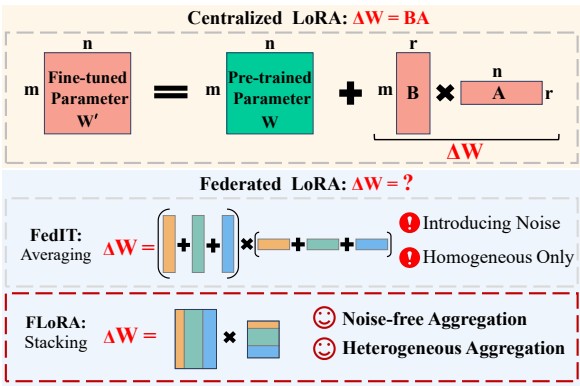

Figure 1: The overview of LoRA, FedIT, and our FLoRA. The top row shows how LoRA updates the model in centralized fine-tuning. The middle and bottom rows show the global model updating strategies in FedIT and our FLoRA respectively.

Furthermore, FLoRA can effectively cater to heterogeneous data and computational resources across clients, where heterogeneous ranks are applied. Our key contributions are summarized as follows:

- We propose FLoRA, a federated fine-tuning algorithm based on LoRA that can perform noise-free aggregation of local LoRA modules. Theoretical analysis shows that FLoRA eliminates the meaningless intermediate term in the global model update, leading to faster convergence and improved performance.
- The proposed stacking mechanism for aggregating LoRA modules supports heterogeneous LoRA ranks across clients, accommodating data and system heterogeneity in realistic settings. This encourages the broader participation of clients with heterogeneous data and resources in federated fine-tuning.
- We use FLoRA to fine-tune LLaMA, Llama2 [21] and TinyLlama [29] on four benchmarks for two downstream tasks. Results show that FLoRA surpasses state-of-the-art methods for both homogeneous and heterogeneous settings.

## 2 Preliminaries

**Fine-tuning LLMs with LoRA.** LoRA [10] uses two decomposed low-rank matrices to represent the update of the target module:

$$\mathbf{W}' = \mathbf{W} + \mathbf{\Delta W} = \mathbf{W} + \mathbf{BA}, \tag{1}$$

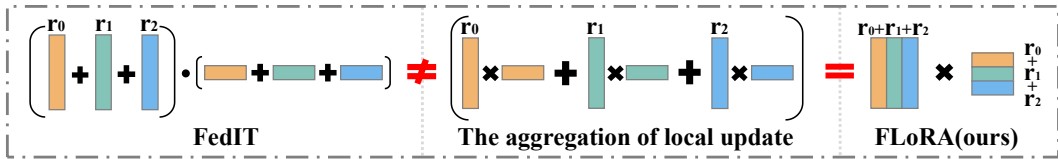

Figure 2: Module stacking in FLoRA is a noise-free aggregation for LoRA, while the module averaging in FedIT cannot accurately aggregate the local updates.

where $\mathbf{W} \in \mathbb{R}^{m \times n}$ and $\mathbf{W}' \in \mathbb{R}^{m \times n}$ denote the pre-trained and fine-tuned parameters of target modules (*e.g.,* attention modules), respectively. $\mathbf{A}$ and $\mathbf{B}$ are low-rank decomposition of $\mathbf{\Delta W}$, where $\mathbf{A} \in \mathbb{R}^{r \times n}, \mathbf{B} \in \mathbb{R}^{m \times r}$, such that $\mathbf{\Delta W} = \mathbf{BA}$ with the identical dimensions as $\mathbf{W}$ and $\mathbf{W}'$. The rank of LoRA, denoted by $r$, is typically significantly smaller than $m$ and $n$, leading to dramatic parameter reduction of $\mathbf{\Delta W}$. During the fine-tuning phase, LoRA optimizes matrices $\mathbf{A}$ and $\mathbf{B}$ instead of directly updating $\mathbf{W}$, thus achieving substantial savings in GPU memory usage. For example, in the context of the LLaMA-7B [21], the original dimension of attention modules is $\mathbf{W} \in \mathbb{R}^{4096 \times 4096}$, setting the LoRA rank to 16 reduces the decomposed matrices to $\mathbf{A} \in \mathbb{R}^{16 \times 4096}$ and $\mathbf{B} \in \mathbb{R}^{4096 \times 16}$. This approach decreases the number of trainable parameters to merely 0.78% of the entire parameter of the pre-trained model, offering a significant GPU memory footprint reduction.

**FedIT: Averaging Homogeneous LoRA.**   The most widely used FL algorithm, *i.e.,* FedAvg [16], aggregates all the local model updates by weighted averaging to update the global model in each communication round:

$$\mathbf{W}' = \mathbf{W} + \sum_{k=1}^{K} p_k \mathbf{\Delta W}_k = \mathbf{W} + \mathbf{\Delta W} \tag{2}$$

where $\mathbf{W}'$ and $\mathbf{W}$ denote the global model parameters before and after a communication round. $\mathbf{\Delta W}_k$ represents the local model update from the $k$-th client, with $p_k$ being the corresponding scaling factor that is typically weighted by the local data size, and $\mathbf{\Delta W}$ represents the global model update.

FedIT [28] directly integrates FedAvg with LoRA to enable federated fine-tuning, where each client fine-tunes LoRA modules with a homogeneous rank. Specifically, the clients download the pre-trained LLM from the server, locally initialize and fine-tune the LoRA modules, and then send the updated LoRA modules to the server. The server updates the global LoRA modules $\mathbf{A}$ and $\mathbf{B}$ by independently applying weighted averaging across all local modules $\mathbf{A}_k$ and $\mathbf{B}_k$:

$$\mathbf{A} = \sum_{k=1}^{K} p_k \mathbf{A}_k, \quad \mathbf{B} = \sum_{i=0}^{K} p_k \mathbf{B}_k. \tag{3}$$

This aggregation of FedIT is almost the same as FedAvg except that only the LoRA modules are trained and communicated. However, such a naive aggregation introduces additional issues for federated fine-tuning. First, each single module $\mathbf{A}$ or $\mathbf{B}$ is not the model update, and only $\mathbf{BA}$ represents the model update. Thus, averaging $\mathbf{A}_k$ and $\mathbf{B}_k$ *independently* to compute the aggregated gradients will introduce noises to the global model update. Here we use a simple example to explain how the noise is generated, and we assume that two clients are applying FedIT to perform federated fine-tuning. In a communication round, the two clients train $\mathbf{A}_0$, $\mathbf{B}_0$ and $\mathbf{A}_1$, $\mathbf{B}_1$ respectively. The local model updates $\mathbf{\Delta W}_0$ and $\mathbf{\Delta W}_1$ are the product of corresponding LoRA modules:

$$\mathbf{\Delta W}_k = \mathbf{B}_k \mathbf{A}_k, k \in \{0, 1\}. \tag{4}$$

According to Equation 2, the expected global model update $\mathbf{\Delta W}$ can be obtained by weighted averaging $\mathbf{\Delta W}_0$ and $\mathbf{\Delta W}_1$:

$$\mathbf{\Delta W} = p_0 \mathbf{\Delta W}_0 + p_1 \mathbf{\Delta W}_1 = p_0 \mathbf{B}_0 \mathbf{A}_0 + p_1 \mathbf{B}_1 \mathbf{A}_1. \tag{5}$$

However, according to Equation 3, FedIT aggregates $\mathbf{A}$ and $\mathbf{B}$ independently:

$$\begin{aligned} \mathbf{\Delta W} = \mathbf{BA} &= (p_0 \mathbf{B}_0 + p_1 \mathbf{B}_1)(p_0 \mathbf{A}_0 + p_1 \mathbf{A}_1) \\ &= p_0^2 \mathbf{B}_0 \mathbf{A}_0 + p_1^2 \mathbf{B}_1 \mathbf{A}_1 + \underline{p_0 p_1 (\mathbf{B}_0 \mathbf{A}_1 + \mathbf{B}_1 \mathbf{A}_0)}. \end{aligned} \tag{6}$$

The global model update in Equation 6 differs from the expected one in Equation 2, mainly due to the underlined intermediate term that is obtained by the cross-product of LoRA modules from

different clients. This intermediate term introduces unexpected noise in the model aggregation. As the number of clients increases, this noisy term becomes much larger than the real global updates, significantly slowing down the fine-tuning progress. In addition, FedIT applies the scaling factor $p_k$ to both $\mathbf{A}_k$ and $\mathbf{B}_k$, resulting in a $p_k^2$ coefficient for the local model update $\mathbf{\Delta W}_k$, exacerbating the error in LoRA aggregation. As Figure 2 illustrates, the averaging algorithm in FedIT is an inaccurate aggregation method, leading to slower convergence and higher computation costs.

The other deficiency of FedIT is that it cannot support aggregation on heterogeneous LoRA modules. The local data in FL may exhibit significant heterogeneity across clients [31; 12]. If a client configures a higher rank than the actual one required by the local data complexity, this may result in overfitting. Conversely, if the rank is too small, it may lack the necessary generalization capacity to effectively learn from the local dataset (Figure 4). Moreover, the heterogeneous computational resource across clients also requires heterogeneous rank deployment, *e.g.,* clients with smaller memory can only afford to train LoRA modules with smaller ranks. AdaLoRA [30] has been proposed to adapt LoRA ranks based on available computation resources. Therefore, deploying heterogeneous ranks across clients is a pressing requirement for accommodation to data and system heterogeneity. However, according to Equation 3, FedIT is only able to aggregate LoRA modules with the homogeneous rank.

## 3 Proposed Method: FLoRA

### 3.1 Stacking-based Noise-free Aggregation

Motivated by the aforementioned problem, we propose a novel aggregation mechanism that accurately computes global model update $\mathbf{\Delta W}$ by aggregating local LoRA modules and effectively supports the heterogeneous LoRA. According to matrix multiplication principles and the model update rule in LoRA (*i.e.,* Equation 1), the element at position $(x, y)$ of the model update $\mathbf{\Delta W}$ is computed as the sum of the products of corresponding elements from the $x$-th column of $\mathbf{B}$ and the $y$-th row of $\mathbf{A}$:

$$\delta_{xy} = \sum_{i=0}^{r} a_{yi} b_{xi}, \tag{7}$$

where $\delta_{xy}$ represents the element at position $(x, y)$ in $\mathbf{\Delta W}$. $a_{yi}, b_{xi}$ are the elements at positions $(y, i)$ and $(x, i)$ in $\mathbf{A}$ and $\mathbf{B}$, respectively. According to Equation 3.1, the model update in LoRA can be expressed as the sum of the products of the corresponding rows of $\mathbf{A}$ and the columns of $\mathbf{B}$.

To illustrate this concept further, let us consider a simplified example where the dimensions of LoRA modules are given by $\mathbf{A} \in \mathbb{R}^{2 \times 3}$ and $\mathbf{B} \in \mathbb{R}^{3 \times 2}$. As described in Equation 8, $\mathbf{A}$ and $\mathbf{B}$ can be decomposed to two sub-matrices with rank $r = 1$, and the product of $\mathbf{A}$ and $\mathbf{B}$ then are computed as the sum of the products of two respective sub-matrices:

$$\mathbf{BA} = \begin{bmatrix} b_{00}, b_{01} \\ b_{10}, b_{11} \\ b_{20}, b_{21} \end{bmatrix} \cdot \begin{bmatrix} a_{00}, a_{10}, a_{20} \\ a_{01}, a_{11}, a_{21} \end{bmatrix} = \begin{bmatrix} b_{00} \\ b_{10} \\ b_{20} \end{bmatrix} \cdot [a_{00}, a_{10}, a_{20}] + \begin{bmatrix} b_{01} \\ b_{11} \\ b_{21} \end{bmatrix} \cdot [a_{01}, a_{11}, a_{21}.] \tag{8}$$

To address the aggregation challenge from an alternative perspective, let us consider the scenario where we have multiple pairs of LoRA modules, $\mathbf{A}_k, \mathbf{B}_k$, optimized by the clients. Each pair satisfies the dimensions $\mathbf{A}_k \in \mathbb{R}^{r_k \times n}$ and $\mathbf{B}_k \in \mathbb{R}^{m \times r_k}$. Similar to Equation 8, the sum of the products of these module pairs is the product of the stacked modules, *i.e.,* $\sum_{k=1}^{K} \mathbf{B}_k \mathbf{A}_k = \mathbf{BA}$, where $\mathbf{B}$ represents the *stacking* of all $\mathbf{B}_k$ modules aligned through dimension $m$ and $\mathbf{A}$ is the *stacking* of all $\mathbf{A}_k$ aligned through dimension $n$. Figure 2 visually illustrates this concept, where the orange, green, and blue rectangles symbolize $\mathbf{A}_k$, $\mathbf{B}_k$, and their respective products. The aggregation of three products mirrors the product of the stacked $\mathbf{B}$ and $\mathbf{A}$ from all $\mathbf{B}_k$ and $\mathbf{A}_k$ pairs trained by clients. This mechanism demonstrates that, in the context of federated fine-tuning, we can achieve a noise-free aggregation of local updates by simply stacking the local LoRA modules. This process also avoids transmitting the full model parameters, thus reducing communication costs.

To facilitate our discussion, we introduce the stacking operation symbolized by "$\oplus$" to denote the module aggregation as depicted in Figure 2. This operation is mathematically defined as:

$$\mathbf{A} = \mathbf{A}_0 \oplus \mathbf{A}_1 \oplus \mathbf{A}_2, \ \mathbf{B} = \mathbf{B}_0 \oplus \mathbf{B}_1 \oplus \mathbf{B}_2,$$
$$\mathbf{A}_k \in \mathbb{R}^{r_k \times n}, \mathbf{A} \in \mathbb{R}^{(r_0 + r_1 + r_2) \times n}, \mathbf{B}_k \in \mathbb{R}^{m \times r_k}, \mathbf{B} \in \mathbb{R}^{m \times (r_0 + r_1 + r_2)}. \tag{9}$$

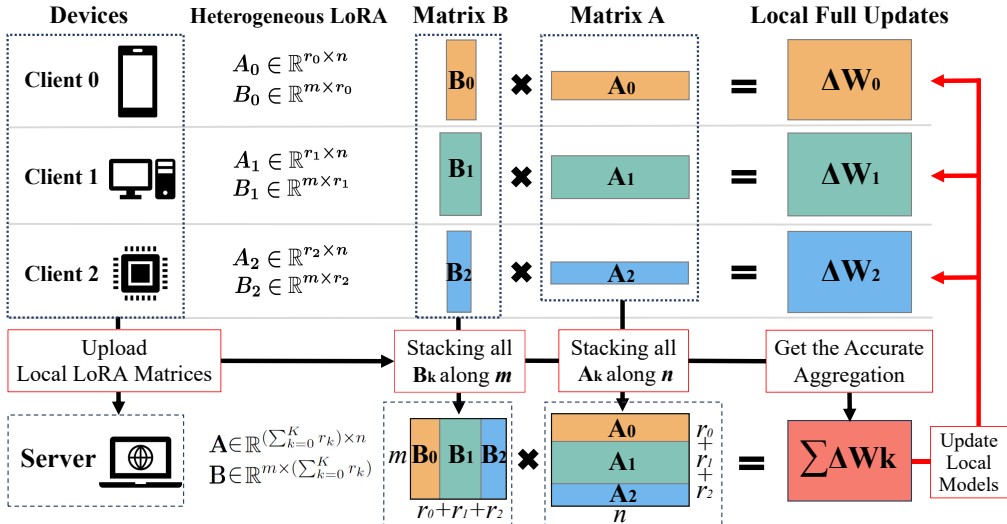

Figure 3: FLoRA workflow. The local LoRA modules are initialized and optimized each round, and stacked by the server to obtain the global LoRA modules. The global modules are then sent back to clients to update local models.

In Equation 9, "$\oplus$" indicates that for $\mathbf{A}$, each subsequent module is vertically stacked below the preceding one, whereas for $\mathbf{B}$, each module is horizontally stacked to the right of the one before it.

We can now formalize our conclusion regarding the aggregation of LoRA modules. The sum of the products of $K$ LoRA module pairs is equivalent to the product of their stacked matrices:

$$\sum_{k=0}^{K} \mathbf{B}_k \mathbf{A}_k = (\mathbf{B}_0 \oplus ... \oplus \mathbf{B}_K)(\mathbf{A}_0 \oplus ... \oplus \mathbf{A}_K) \tag{10}$$

This foundational principle will guide the design of FLoRA, as it allows for the efficient and effective aggregation of local updates without the transmission of entire model parameters.

### 3.2  FLoRA: Stacking-based Federated Fine-tuning for Heterogeneous LoRA

The stacking-based aggregation facilitates not only the accurate aggregation of LoRA modules but also inherently supports the heterogeneous LoRA ranks. This approach imposes no constraints on the ranks of each local LoRA module as long as each client fine-tunes the same pre-trained model, *i.e.,* they share the same dimension $m$ and $n$.

By employing the stacking-based aggregation mechanism, we introduce FLoRA, an approach designed to facilitate federated fine-tuning of LLMs with heterogeneous LoRA. Let us use a concrete example to illustrate the key steps of applying FLoRA, where $K$ heterogeneous clients are involved in fine-tuning an LLM, and the pre-trained parameters are denoted by $\mathbf{W}$.

**Initialization.**  The server first disseminates the pre-trained model parameters $\mathbf{W}$ to all $K$ clients. Then, the clients initialize their local LoRA modules based on the complexity of local data and available local resources. The adaptation of LoRA ranks is beyond the scope of this paper, but existing work like AdaLoRA [30] can facilitate the rank adjustment.

**Local Fine-tuning.**  Following initialization, the clients train their local LoRA modules with the local data for several iterations. Then, the clients send the local LoRA modules back to the server. Note that the clients initialize local LoRA modules *each* round before local fine-tuning.

**Stacking-based LoRA Aggregation.**  Upon receiving the heterogeneous LoRA modules from participating clients, the server proceeds to aggregate them by stacking all $\mathbf{B}_k$ and $\mathbf{A}_k$ according to Equation 10, resulting in the global $\mathbf{A} \in \mathbb{R}^{(\sum_{k=0}^{K} r_k) \times n}$ and $\mathbf{B} \in \mathbb{R}^{m \times (\sum_{k=0}^{K} r_k)}$. The aggregation

process of FLORA can be described as follows:

$$\mathbf{A} = p_0\mathbf{A}_0 \oplus p_1\mathbf{A}_1 \oplus ... \oplus p_K\mathbf{A}_K, \quad \mathbf{B} = \mathbf{B}_0 \oplus \mathbf{B}_1 \oplus \mathbf{B}_2 \oplus ... \oplus \mathbf{B}_K$$
$$\mathbf{A}_k \in \mathbb{R}^{r_k \times n}, \mathbf{B}_k \in \mathbb{R}^{m \times r_k}, \quad \mathbf{A} \in \mathbb{R}^{(\sum_{k=0}^K r_k) \times n}, \mathbf{B} \in \mathbb{R}^{m \times (\sum_{k=0}^K r_k)}, \tag{11}$$

where $p_k$ represents the scaling factor for each local update, determined by the relative size of the local data to the global data:

$$p_k = \frac{len(D_k)}{len(\sum_{k=0}^K D_k)}. \tag{12}$$

Note that the scaling factor $p_k$ should be only applied to one of $\mathbf{A}_k$ and $\mathbf{B}_k$ to avoid squaring the factor in the final model update $\mathbf{BA}$. This method ensures a noise-free aggregation mechanism as described in Equation 10.

**Update Local Models.**   After each round of noise-free aggregation, the server redistributes the updated global LoRA modules $\mathbf{A}$ and $\mathbf{B}$ back to the clients. The clients then proceed to update the local models using $\mathbf{BA}$ and continue the fine-tuning. Using the stacking approach, the dimensions of updated global LoRA modules $\mathbf{A}$ and $\mathbf{B}$ are larger than those of FedIT, potentially leading to larger communication overhead in each round. However, empirical observations indicate that federated fine-tuning typically requires only a limited number of communication rounds to achieve satisfactory results, as detailed in Section 4. In addition, it is important to note that the LoRA modules $\mathbf{A}$ and $\mathbf{B}$ constitute a small fraction of the overall size of the pre-trained model, which is distributed to clients during the initialization phase. Thus, the additional communication overhead of the stacking approach is negligible and does not significantly impact the efficiency of federated fine-tuning.

## 4   Experiments

The key features of FLORA are (i) noise-free aggregation and (ii) support for heterogeneous LoRA modules. In this section, we verify these key features across various LLM fine-tuning tasks. We first study the performance of FLORA and compare it against FedIT under homogeneous settings to demonstrate the advantages of noise-free aggregation [28]. Then, we examine performance in a synthetic heterogeneous setup and compare FLORAwith a vanilla *zero-padding* method. Finally, we conduct ablation studies on the scaling factor, the heterogeneity of LoRA ranks, and the extra communication overhead of FLoRA.

### 4.1   Experiment Setup

**Models, Datasets and Experiment Settings.** We employ three Llama-based models with different scales in our experiments: TinyLlama with 1.1 billion parameters [29], and the 7 billion parameter versions of Llama [21] and Llama2 [22], evaluating FLORA across different model capacities. Following the configurations in the original LoRA paper [10], the LoRA modules are applied to the self-attention layers only.

We use the Databricks-dolly-15k [28] instruction dataset, Alpaca dataset [19], and Wizard dataset [14] for the question-answering (QA) task, and Wizard and ShareGPT for the chat assistant task. We evaluate the federated fine-tuned models on MMLU [8] for the QA task and MT-bench [32] for the chat assistant task, respectively. We sample 10 clients uniformly at random following the non-IID setting in FedIT [28]. The other experimental configurations are elaborated in Appendix A.

**Baselines.**   We compare FLORA with four baselines. (1) **FedIT:** It is the SOTA federated fine-tuning method [28] that integrates LoRA with FedAvg. We only apply FedIT to homogeneous LoRA experiments as it does not support heterogeneous LoRA. (2) **Zero-Padding:** It is an approach that enables FedIT to support heterogeneous LoRA [3]. It extends all the heterogeneous local ranks to the maximum rank among the clients and pads their remaining parts by 0. (3) **Centralized Fine-tuning:** we compare FLORA with centralized LoRA with the same hyperparameters and configurations. (4) **Standalone:** the client fine-tunes the pre-trained model locally without federations.

Table 1: Comparison of FLoRA with baselines on MMLU and MT-bench. "Homo" represents the settings with homogeneous LoRA ranks, and "Heter" denotes those with heterogeneous LoRA ranks.

| Foundation model | Strategy | Fine-tuning algorithm | MMLU | | | MT-bench | |
| --- | --- | --- | --- | --- | --- | --- | --- |
| | | | Dolly | Alpaca | Wizard | Wizard | ShareGPT |
| TinyLlama | Centralized | LoRA | 27.99 | 28.03 | 29.13 | 2.34 | 2.79 |
| | Homo | FedIT | 16.35 | 30.02 | 42.51 | 2.92 | 2.55 |
| | | FLoRA | **30.80** | **31.92** | **43.87** | **3.13** | **2.77** |
| | Heter | Zero-Padding | 15.76 | 29.56 | 40.79 | 1.56 | 1.29 |
| | | FLoRA | **18.45** | **29.69** | **41.48** | **3.14** | **2.71** |
| Llama | Centralized | LoRA | 35.91 | 29.18 | 31.68 | 4.38 | 3.99 |
| | Homo | FedIT | 29.67 | 29.41 | 33.43 | 3.07 | 3.73 |
| | | FLoRA | **30.99** | **29.85** | **34.26** | **4.21** | **3.93** |
| | Heter | Zero-Padding | 26.46 | 7.97 | 26.98 | 3.51 | 3.26 |
| | | FLoRA | **28.50** | **29.54** | **27.91** | **4.14** | **3.64** |

## 4.2 Experiment Results

**Homogeneous LoRA.** We first evaluate the performance of FLoRA with homogeneous LoRA. Specifically, all the clients share the identical LoRA rank of 16. As Table 1 depicts, FLoRA achieves consistently better performance than FedIT across all the evaluated models and tasks. This is evident in the MT-bench scores for both TinyLlama and Llama models, where FLoRA's performance exceeds that of FedIT by at least 0.2. A notable example is the MT-bench score for the Llama model fine-tuned with Wizard dataset, where FLoRA scores 4.21, surpassing FedIT's 3.07. On the MMLU test set, FLoRA outperforms FedIT in all the settings. For example, considering the TinyLlama model fine-tuned with Dolly, FLoRA nearly doubles the accuracy achieved by FedIT. While FedIT occasionally matches the performance of FLoRA, as observed with the Alpaca dataset on MMLU, the performance gap is marginal. Interestingly, in several scenarios, the performance of FLoRA not only outpaces FedIT but also exceeds the performance achieved by the centralized fine-tuning. This phenomenon, observed in the TinyLlama model fine-tuned with the Alpaca and Wizard datasets, suggests that the smaller data volume on clients for federated fine-tuning may help mitigate overfitting, thereby enhancing model generalization. The experiment results of the Llama2 model are presented in Appendix A, which reveal the same trend as that in TinyLlama and Llama. The consistent observations across the three models demonstrate that FLoRA consistently outperforms FedIT in the homogeneous LoRA setting.

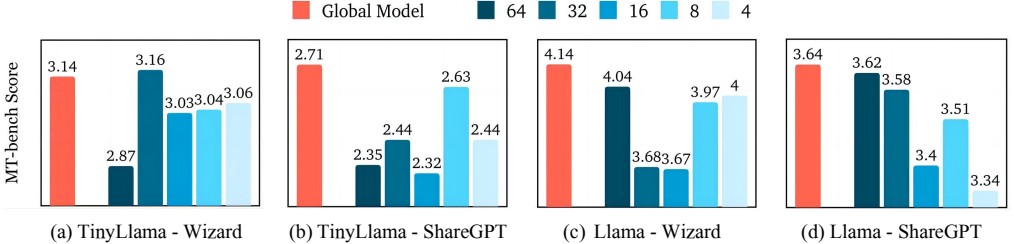

(a) TinyLlama - Wizard     (b) TinyLlama - ShareGPT     (c) Llama - Wizard     (d) Llama - ShareGPT

Figure 4: Standalone experiment results. The red bars represent the global model performance and the blue bars represent the local model performance with varying LoRA ranks.

**Heterogeneous LoRA.** Compared with FedIT, a distinctive strength of FLoRA lies in its inherent capability to accommodate heterogeneous LoRA configurations. In the heterogeneous LoRA settings, we apply varied local LoRA ranks, *i.e.,* [64, 32, 16, 16, 8, 8, 4, 4, 4, 4], to 10 clients, simulating a realistic scenario where clients have heterogeneous computational resources. As Table 1 and Table 4 illustrate, FLoRA not only adapts to heterogeneous ranks without performance degradation but also maintains consistency with the results observed in most homogeneous settings. This contrasts sharply with the performance of FedIT, where the application of zero-padding significantly degrades its performance on MMLU and MT-bench. It reveals that zero-padding exacerbates FedIT's inherent noise issues in the aggregation process, posing significant challenges in managing fine-tuning performance. For example, by applying the zero-padding method, the MMLU accuracy of Llama model fine-tuned with Alpaca dataset dramatically drops to 7.97%. The results demonstrate that FLoRA

not only accommodates heterogeneous LoRA ranks effectively but also sustains robust training performance compared to baseline methods. It efficiently facilitates the participation of devices with varied computational capacities in heterogeneous federated fine-tuning tasks. Additionally, FLoRA can be seamlessly integrated with AdaLoRA [30], which dynamically adjusts the LoRA rank on the clients, the results are presented in Appendix A.

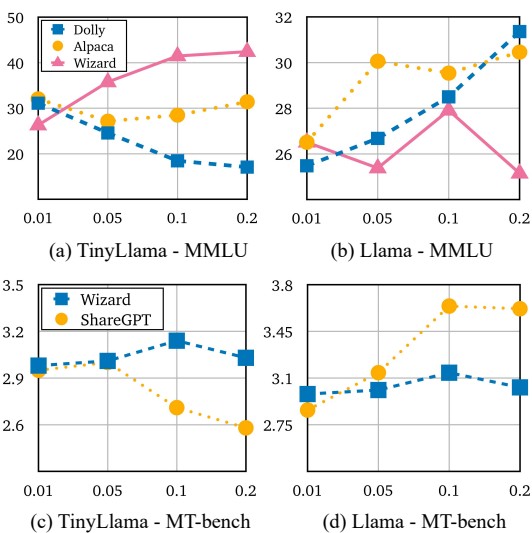

(a) TinyLlama - MMLU  (b) Llama - MMLU

(c) TinyLlama - MT-bench  (d) Llama - MT-bench

Figure 5: The impact of the scaling factor on FLoRA. The x-axis is the scaling factor, and the y-axis represents the MMLU accuracy for (a)-(b) and the MT-bench score for (c)-(d). The results of Llama2 are in Appendix A.

**The Impact of Scaling Factor.** The scaling factor, denoted as $p_k$ in Equation 12, plays a pivotal role in the efficacy of FL [23]. We conduct experiments investigating how varying scaling factors influence the performance of FLoRA. Given that the default scaling factor is set to 0.1 for all clients, assuming 10 clients with equal local dataset sizes as per Equation 12, we explored the effects of alternative scaling factors, namely 0.01, 0.05, and 0.2. The results are summarized in Figure 5. The results do not reveal a clear pattern or optimal scaling factor for federated fine-tuning across different settings. The efficacy of a specific scaling factor appears to be contingent upon the dataset, task, and model in use. For example, when fine-tuning TinyLlama on the Dolly dataset, a lower scaling factor of 0.01 yields the highest accuracy, significantly outperforming the 0.1 and 0.2 scaling factors. Conversely, the model fine-tuned on Wizard dataset demonstrates a preference for a higher scaling factor of 0.2, achieving the best performance, whereas the lowest scaling factor of 0.01 was the least effective. In the case of the Llama model, larger scaling factors consistently facilitated better fine-tuning performance. Applying FLoRA to Dolly and Alpaca shows the optimal performance with a scaling factor of 0.2. These observations suggest that the choice of an appropriate scaling factor is highly dependent on specific datasets and model characteristics, underscoring the necessity for a tailored approach in federated fine-tuning.

**The Impact of Heterogeneous LoRA Ranks.** Although the above results demonstrate FLoRA effectively enables the federated fine-tuning with heterogeneous LoRA, it is worth further investigating how the federated fine-tuning improves the local models with various ranks. Motivated by this, we evaluate MT-bench scores for local models with LoRA ranks of 64, 32, 16, 8, and 4, presenting the results in Figure 4. Global model scores are shown in red bars, while local models are in blue, with deeper shades indicating higher ranks. The results show that the global model outperforms all local models, except for a case with the TinyLlama model fine-tuned on the Wizard dataset, where the client with rank 32 slightly exceeds the global model. This demonstrates FLoRA's ability to synthesize knowledge from diverse clients effectively.

Regarding the LoRA rank's impact, a rank of 8 consistently yields strong performance across various models and datasets. However, performance diverges at extreme ranks; for instance, the TinyLlama model fine-tuned on Wizzard with the LoRA rank of 64 underperforms the ones with smaller ranks, but the Llama model with the rank of 64 excels the counterparts with smaller ranks. This also demonstrates the heterogeneous rank deployment across clients is a realistic setting. These observations suggest a potential positive correlation between optimal LoRA rank and model capacity, motivating further exploration in future research.

# 5 Discussion

**The Communication Overhead of FLoRA.** As discussed in Section 3, the server needs to send global LoRA modules to the clients in FLoRA, potentially raising concerns about increased commu-

nication overhead. To quantify this, we compare the communicated parameters of full fine-tuning, FedIT, and FLoRA over three communication rounds.

As Figure 6 shows, although FLoRA transmits slightly more parameters than FedIT, it still significantly reduces the overhead compared to full fine-tuning. This is due to the fact that the primary communication load in federated fine-tuning, especially with large models, is the initial full model parameter transmission. Subsequent rounds primarily involve smaller updates (e.g., LoRA matrices). Thus, even though FLoRA introduces additional communication for these updates, the overall impact on total communication costs remains marginal, making it comparable to FedIT's costs. Despite the minor communication increase compared to FedIT, FLoRA enhances fine-tuning effectiveness and supports heterogeneous LoRA ranks, making it a preferable solution in federated fine-tuning.

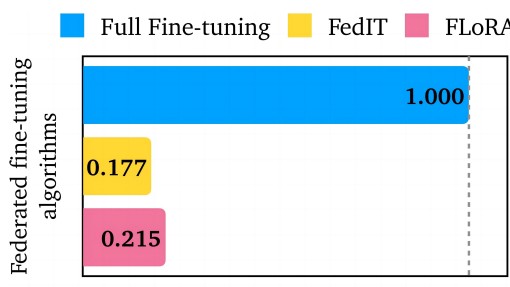

Ratio of communicated parameter # to full fine-tuning — 3 communication rounds

Figure 6: The ratio of communicated parameter numbers to full fine-tuning.

**The Privacy Preservation of FLoRA.** The requirement of FLoRA to stack the LoRA modules uploaded by all clients introduces a potential privacy concern, as malicious clients might infer the LoRA matrices of other clients through the global LoRA modules sent from the server. To address this issue, we split all the local LoRA modules into sub-modules with rank=1 and then stack the sub-modules together in random order. This approach prevents malicious clients from recovering the local LoRA modules from other clients. In addition, FLoRA is also compatible with standard privacy mechanisms such as encryption [2] and differential privacy (DP) [25], aligning it with the privacy-preserving nature of FL.

# 6    Related Work

**Parameter-efficient Fine-tuning of LLMs.** Parameter-efficient fine-tuning (PEFT) aims to reduce the number of trainable parameters. BitFit [26] fine-tunes only the biases while achieving similar accuracy with full fine-tuning. Other works such as [9] and [17] apply transfer learning that adds pre-trained adapter layers between transformer blocks. LoRA [10] adopts the product of two low-rank matrices to represent the gradient in full fine-tuning, which achieves memory-efficient fine-tuning. AdaLoRA [30] optimizes LoRA by adaptively allocating the parameter budget, which enhances the flexibility of LoRA. There are also many works regarding optimizing LoRA in various aspects [5; 6; 15].

**Federated Fine-tuning of LLMs.** Federated fine-tuning aims to extract knowledge from multiple on-device datasets while preserving data privacy. FedIT [28] leverages the FL framework for fine-tuning LLMs. It uses LoRA as the local fine-tuning strategy. However, concerns related to the deficiency in supporting heterogeneous LoRA limit its utilization. [3] tries to solve this problem by zero-padding the local LoRA modules. However, this padding process causes additional computing overhead. Besides, it separately averages **A** and **B** modules, introducing noise to the global model.

# 7    Conclusion

In this work, we identified the limitations in current federated fine-tuning methods (*e.g.,* FedIT), and the challenges of applying federated fine-tuning in realistic settings, *i.e.,* the heterogeneous LoRA ranks across clients. To overcome these practical challenges and broaden the applicability of federated fine-tuning, we introduced FLoRA to enable the accurate aggregation on heterogeneous LoRA modules using the proposed stack-based LoRA aggregation mechanism. Our extensive experiments demonstrate that FLoRA outperforms the SOTA method in both homogeneous and heterogeneous LoRA settings. Moreover, our inspiring results provide valuable insights for future research in the federated fine-tuning of large language models in a lightweight and accurate manner.

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

# A  Additional Experiments and Setup Details

## A.1  Environments, Datasets and Metric

**Computer Resources.** We use a 256GB AMD EPYC 7763 64-Core Processor on Linux v4.18.0 to run the experiments. For LoRA fine-tuning on all the models, we use 4 NVIDIA RTX A6000 GPUs.

**Dolly dataset.** The Dolly dataset is an open-source dataset with 15k text samples generated by Databricks employees. The topics include brainstorming, classification, closed QA, generation, information extraction, open QA, and summarization [28].

**Alpaca dataset.** The Alpaca dataset contains 52K instruction-following data used for fine-tuning the Alpaca model [19]. This dataset is believed to be diverse enough for fine-tuning LLMs.

**Wizard dataset.** The Wizard dataset we use is the training data of the WizardLM model. It includes 70k pairs of instructions and outputs. The Wizard dataset generally features more complex instructions compared to the other datasets. Its fine-tuning results are typically better, which has been confirmed by our experiments, especially those evaluated by the MT-bench scores.

**ShareGPT dataset.** The ShareGPT dataset is a collection of approximately 52,000 conversations scraped via the ShareGPT API. The conversations in ShareGPT include both user prompts and responses from ChatGPT. In our experiments, we split the conversation dataset into question-answering pairs.

**MMLU test set.** The MMLU dataset is a widely used question-and-answer dataset in LLM fine-tuning. It has 14,024 questions in 57 different subjects, which can evaluate the logical reasoning capabilities of LLMs. We selected 1444 samples from the dataset for a quick and comprehensive evaluation.

**MT-bench evaluation.** MT-bench is a set of challenging multi-turn open-ended questions for evaluating chat assistants [32]. It evaluates the performance of LLMs by using the GPT-4 API to score the LLM-generated conversations. LLMs that behave more like GPT-4 will receive higher scores.

## A.2  Hyperparameter Details

In all our experiments, the learning rate of fine-tuning is set to 0.0003; the batch size is 128 and the micro batch size is 16. Due to the large dataset and model sizes selected, federated fine-tuning consumes significant computational resources and time. Therefore, we opted for fewer fine-tuning rounds (even just one round) to ensure that we could observe enough data. Additionally, the MMLU dataset is prone to overfitting on these large datasets, resulting in a decrease in accuracy. Therefore, fewer training rounds ensure the effectiveness of the observed phenomena. Table 2 shows the fine-tuning rounds and local epochs we selected.

Table 2: The communication rounds and local epochs on each experiment setting. The Rounds column represents the number of communication rounds and the Epochs column represents the number of local fine-tuning epochs in each round.

| Foundation | Datasets | Rounds | Epochs |
|---|---|---|---|
| TinyLlama | Dolly | 3 | 1 |
| | Alpaca | 3 | 1 |
| | Wizard | 3 | 1 |
| | ShareGPT | 1 | 1 |
| Llama | Dolly | 3 | 3 |
| | Alpaca | 3 | 3 |
| | Wizard | 1 | 1 |
| | ShareGPT | 1 | 1 |
| Llama2 | Wizard | 1 | 1 |
| | ShareGPT | 1 | 1 |

Table 3: The performance of FLoRA + AdaLoRA. AdaLoRA can reduce the rank while preserving the fine-tuning effectiveness.

| Foundation model | Fine-tuning algorithm | Sum of local ranks | MT-bench score |
|---|---|---|---|
| TinyLlama | FLoRA | 160 | 3.13 |
| | FLoRA+AdaLoRA | 120 | 3.14 |
| Llama | FLoRA | 160 | 4.21 |
| | FLoRA+AdaLoRA | 131 | 4.10 |
| Llama2 | FLoRA | 160 | 4.17 |
| | FLoRA+AdaLoRA | 140 | 4.25 |

Table 4: Compare FLoRA with baselines in Llama2.

| Strategy | Fine-tuning algorithm | Wizard | ShareGPT |
|---|---|---|---|
| Centralized | LoRA | 4.24 | 3.99 |
| Homo | FedIT | 4.03 | 3.87 |
| | FLoRA | 4.22 | 3.96 |
| Heter | Zero-Padding | 4.01 | 3.70 |
| | FLoRA | 4.17 | 3.91 |

## A.3 Supplementary Experiment Results

**Integrating FLoRA with AdaLoRA** All the observations about the impact of rank on the model performance, despite being influenced by data heterogeneity, still manage to reveal the importance of selecting an appropriate LoRA rank for a specific task. Thus, some algorithms such as AdaLoRA [30] are designed to adaptively adjust the LoRA rank to optimize the model performance and save computational resources. With our support for heterogeneous LoRA, we can flexibly utilize AdaLoRA with adaptive LoRA ranks. We conducted corresponding experiments to demonstrate that we can use AdaLoRA to further improve the efficiency of federated fine-tuning. We implement AdaLoRA on each client to adjust LoRA modules during local fine-tuning. The results are shown in Table 3. The "Sum of local ranks" column means the sum of all local LoRA rank values *after* fine-tuning. Since our FLoRA does not adjust the rank, its value is 160, the same as the initial value. On the other hand, AdaLoRA dynamically adjusts the rank to maximize training effectiveness and minimize rank values to save resources. From Table Table 3, we can see that AdaLoRA on TinyLlama and Llama reduced the sum of local ranks to 120 and 131 from 160 respectively. We further conclude that FLoRA+AdaLoRA can further reduce the trainable parameter count while ensuring comparable or even improved performance compared to simply using LoRA on the clients. Our support for such rank adaptation further demonstrates the effectiveness and applicability of the FLoRA approach.

**Experiment results of Llama2.** Due to the inherently strong performance of Llama2, the improvement in the QA dataset is not significant. Therefore, we fine-tuned Llama2 using the Wizard and ShareGPT datasets. Overall, Llama2 exhibits similar experimental results to Tinyllama and Llama. Table 4 shows the comparison between FLoRA and our baselines. In the homogeneous and heterogeneous settings, the MT-bench scores of Wizard and ShareGPT all surpass those in FedIT and Zero-Padding. As for the impact of scaling factors in Figure 7, Llama2 has a similar trend to the Llama-7b model, in which higher scaling factors exhibit better fine-tuning performance.

## B   Convergence Analysis

In this section, we demonstrate the convergence of FLoRA following the standard convergence analysis in [12]. The FedAvg algorithm exhibits convergence to the global optimum at a rate of $O(1/T)$ for non-IID (independent and identically distributed) data under full client participation. This convergence is based on four assumptions mentioned in [12]:

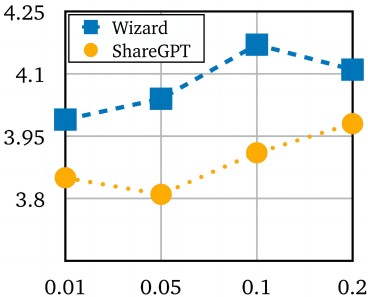

Figure 7: The impact of scaling factor on Llama2 model.

**Assumption 1.** Each local objective function is L- smooth, that is, for all $x$ and $y$, $F_k(x) \leq F_k(y) + (x-y)^T \nabla F_k(y) + \frac{L}{2}\|x-y\|_2^2$

**Assumption 2.** Each local objective function is $\mu$ - strongly convex that is, for all $x$ and $y$, $F_k(x) \geq F_k(y) + (x-y)^T \nabla F_k(y) + \frac{\mu}{2}\|x-y\|_2^2$

**Assumption 3.** The variance of stochastic gradients in each client is bounded: $\mathbb{E}\|\nabla F_k(W_k^{(t)}, \xi_k^{(t)}) - \Delta W_k^{(t)}\|^2 \leq \sigma_k^2$ for $k = 1, ..., K$, where $\xi_k^{(t)}$ is the subset of training data randomly sampled from $k$-th client.

**Assumption 4.** The expected squared norm of stochastic gradients is uniformly bounded: $\mathbb{E}\|\nabla F_k(W_k^{(t)}, \xi_k^{(t)})\|^2 \leq G^2$ for all $k = 1, ..., K$ and $t = 1, ..., T$, where $\xi_k^{(t)}$ is the subset of training data randomly sampled from $k$-th client.

For the convergence analysis of FLoRA, we introduce an additional assumption 5 tailored to the specific dynamics of LoRA fine-tuning and its relation to traditional SGD-based full fine-tuning:

**Assumption 5.** (Unbiased LoRA Gradient). The updates applied to LoRA modules by each client serve as unbiased estimators of the gradient that would be directly computed on the base model through SGD: $\mathbf{B}_k^{(t+1)}\mathbf{A}_k^{(t+1)} - \mathbf{B}_k^{(t)}\mathbf{A}_k^{(t)} = \eta^{(t)}\nabla F_k(\mathbf{W}_k^{(t)}|\xi_k^{(t)})$. Note that we define the model parameter in $t$-th round by $\mathbf{W}^{(t)}$.

**Theorem 1.** Based on Assumptions 1-5, we choose $k = \frac{L}{\mu}$, $\gamma = \max\{8k, E\}$. The local learning rate $\frac{\alpha_k}{r_k} = \frac{2}{\mu(\gamma+t)}$. Then, we can deduce that the expectation of the fine-tuning error in FLoRA can be bounded by:

$$\delta^{(T)} \leq \frac{2k}{\gamma+T}(\frac{M}{\mu} + 2L\|\mathbf{W}^{(1)} - \mathbf{W}^*\|^2), \tag{13}$$

where $\delta^{(T)}$ is the fine-tuning error in $T$-th round. $\delta^{(T)}$ and $M$ are defined as follows:

$$\delta^{(T)} = \mathbb{E}[F(\mathbf{w}^{(T-1)} + \mathbf{B}^{(T)}\mathbf{A}^{(T)})] - F^*,$$

$$M = \sum_{k=1}^{K} p_k^2 \sigma_k^2 + 6L\Gamma + 8(E-1)^2 G^2. \tag{14}$$

where $L, \mu, \sigma_k$, and $G$ are defined by the assumptions 1-4. $\Gamma$ is defined by $\Gamma = F* - \sum_{k=1}^{K} p_k F_k^*$ for quantifying the degree of non-iid. This theorem posits that as the number of rounds $T$ approaches infinity, the expectation of the fine-tuning error $\delta^{(T)}$ converges to zero. In contrast, FedIT deviates from the FedAvg model updating rule as depicted in Equation 2, introducing non-gradient noises through its averaging process. Therefore, it fails to achieve convergence at the rate of $O(1/T)$ While this deviation does not invalidate FedIT's utility in federated fine-tuning, it significantly impairs its convergence rate and overall effectiveness.

## C  Limitation

Our approach has the limitation that the server sends the stacked LoRA modules to the client, thereby increasing the communication costs. We discussed this limitation both theoretically and

experimentally in Section 3 and Section 4, respectively. We believe that the increase in communication overhead is acceptable under the premise of improving fine-tuning effectiveness and accelerating convergence. In addition, due to constraints on computational resources and time, we only utilized Llama models in the experimental section. We aim to observe experimental phenomena of different types of LLM federated fine-tuning in future research and derive more general principles.

