# OpenReview forum: "FLoRA: Federated Fine-Tuning Large Language Models with Heterogeneous Low-Rank Adaptations"
_NeurIPS.cc/2024/Conference — NeurIPS 2024 poster_

### Official Review · Reviewer_G8j8 · 2024-07-08

**Soundness:** 3
**Presentation:** 3
**Contribution:** 3
**Rating:** 7
**Confidence:** 4

**Summary:**

This paper proposes a novel aggregation algorithm FLoRA for low-rank federated fine-tuning. Compared to prior works, FLoRA stacks the LoRA matrices together rather than averaging them, thus avoid the mathematical errors during the aggregation process on the server. The paper provides experiments compared with baselines and discusses about the efficiency and privacy preservation of FLoRA. The paper provides convergence analysis of FLoRA.

**Strengths:**

(1) This paper identifies and addresses an issue in LoRA federated learning algorithm from a unique perspective. The mathematical analysis provided in the paper is straightforward and convincing. This work can also provide intuition to other works related to LoRA-based merging.
(2) The paper is well-written and easy to follow.
(3) The paper provides sufficient experiments and analysis.

**Weaknesses:**

(1) The paper misses some related works. For example, [1] also talks about LoRA in federated fine-tuning and optimize its efficiency and accuracy. [1] and other possible related works should be considered and added to the paper.
(2) The experiment results presented in Figure 4 show that some single client perform better than the server, which needs further explanation in the paper. Additionally, the subtitle 'The Impact of Heterogeneous...' here does not match the title 'Standalone...' of Figure 4, which needs further polishing.
(3) The paper should explain why it uses Llama rather than SOTA models (e.g., Llama-3).

[1] Sun, Y., Li, Z., Li, Y., & Ding, B. (2024). Improving LoRA in privacy-preserving federated learning. arXiv preprint arXiv:2403.12313.

**Questions:**

Please see the weaknesses. The author might need to explain why they are using Llama rather than stronger models like Llama-2 or Llama-3.

**Limitations:**

The limiations are discussed.

---

> ### Author Rebuttal · Authors · 2024-08-06
>
> We greatly appreciate the reviewer's recognition of the contributions of our paper. Regarding the weaknesses raised, we address all the concerns in detail below.
>
> >W1 The paper misses some related works. For example, [1] also talks about LoRA in federated fine-tuning and optimize its efficiency and accuracy. [1] and other possible related works should be considered and added to the paper.
>
> [1] is a quite relevant paper also discussing the error of simply averaging LoRA, and focuses on the privacy issues in federated learning to aggregate LoRA modules. We will add a detailed discussion regarding the methods and insights of [1] in our future version.
>
> >W2 The experiment results presented in Figure 4 show that some single client perform better than the server, which needs further explanation in the paper. Additionally, the subtitle 'The Impact of Heterogeneous...' here does not match the title 'Standalone...' of Figure 4, which needs further polishing.
>
> In our m=10 experiments, the training dataset is randomly divided into ten parts and distributed to clients for local training, resulting in an uneven quality distribution of the local datasets. Many studies have pointed out that datasets often contain a large amount of useless or even harmful data. As a result, it is acceptable that some clients may train local models that perform better than the global model due to receiving higher-quality or more suitable data for the test set.
> We will change the title of Figure 4 to make it clearer and more accurate.
>
> >W3 The paper should explain why it uses Llama rather than SOTA models (e.g., Llama-3).
>
> In our experiments, we observed that state-of-the-art (SOTA) models are not very sensitive to common fine-tuning datasets. Some models show no performance improvement after fine-tuning or even experience performance declines. For instance, after 3 epochs of fine-tuning on MMLU, we tested Llama-2-7b and obtained the following results:
>
> ==========================
>
> $\quad$ $\quad$ $\quad$   $\quad$   Before      $\quad$  $\quad$     After
>
> —-------------------------------------------
>
> MMLU  $\ $  $\ $  $\quad$    45.80   $\quad$  $\quad$   45.19
>
> HellaSwag $\ $ $\ $58.69 $\quad$   $\quad$59.47
>
> ==========================
>
> We believe this occurs because stronger models like Llama-2-7b have already converged on some common benchmarks, making further fine-tuning less effective in increasing accuracy. The performance of the Llama-3 series is also similar to this. To better demonstrate the effectiveness of fine-tuning, and to provide a clearer comparison between FLoRA and baseline methods, we chose models like Llama-7b where the benefits of fine-tuning are more evident.
>
>
> Still, following the advice of reviewer 7eRV, we extend our experiments to other series of models, here are some results:
>
> ===========================================
>
> Training Set    $\quad$   $\quad$    $\quad$     Gemma-2b  $\quad$    Gemma-7b
>
> —------------------------------------------------------------------------
>
> MMLU   $\quad$   $\quad$  FLoRA    $\quad$ $\quad$  3.40      $\quad$   $\quad$  $\quad$  5.89
>
> $\quad$  $\quad$ $\quad$  $\quad$ $\ $ FedIT     $\quad$  $ $ $\quad$ 3.33  $\quad$   $\quad$ $\quad$  5.72
>
> Wizard   $\quad$   $\ $ $\ $  FLoRA    $\quad$  $\quad$  3.54   $\quad$   $\quad$ $\quad$  6.65
>
> $\quad$  $\quad$ $\quad$  $\quad$ $\ $ FedIT    $\quad$ $ $ $\quad$  3.44    $\quad$    $\quad$  $\quad$ 6.65
>
> ===========================================

---

> > ### Comment · Reviewer_G8j8 · 2024-08-10
> >
> > Thank you for addressing the concerns raised during my review process, and I have also gone through all the comments from other reviewers. I agree with Reviewer 7eRV on the contribution of this work, the proposed method is simple yet effective. Our community indeed benefits from straightforward methods that address realistic problems. The simplicity of this work is not a drawback; rather, it facilitates broader use and adoption by the entire community.
> >
> > The authors have successfully identified and rectified the mathematical errors present in current methods with their approach. Therefore, I believe this paper should be accepted by NeurIPS 2024 to attract broader attention from the community. More researchers need to explore the proposed method and potentially develop novel solutions to address the identified problem.
> >
> > Based on the points mentioned above, I have decided to raise my score.

---

### Official Review · Reviewer_7eRV · 2024-07-11

**Soundness:** 3
**Presentation:** 3
**Contribution:** 3
**Rating:** 7
**Confidence:** 3

**Summary:**

Previous methods using Low-Rank Adaptation (LoRA) for efficient federated fine-tuning may have led to mathematically inaccurate aggregation noise, reducing the effectiveness of fine-tuning and being unable to handle heterogeneous LoRA. The authors analyzed the mathematical inaccuracies in LoRA aggregation in existing federated fine-tuning methods and proposed a new stack-based aggregation method that supports federated fine-tuning across client machines with heterogeneous LoRA adapters. Extensive experiments demonstrate the superior performance of FLORA in homogeneous and heterogeneous environments.

**Strengths:**

1. The proposed method is simple, effective, and easy to implement.
2. The stacking mechanism for aggregating LoRA modules supports heterogeneous LoRA ranks across clients, which has broader application scenario.
3. The experiments utilized various models and benchmarks, and the results validated their effectiveness.

**Weaknesses:**

1. Writing needs to be calibrated. The sentences in line 64 and line 70 are repeated.
2.The discussion on accelerating convergence is not indexed in the main text (Appendix B) and lacks experimental validation.
3. More discussion may be needed on the advantages of FLoRA compared to the implementation of 'the aggregation of local update' in Figure 2.
4. The foundation models chosen for the experiment are all from the Llama series. Are there other types of foundation models that can be used for validation?

**Questions:**

see the weaknesses above

**Limitations:**

see the weaknesses above

---

> ### Author Rebuttal · Authors · 2024-08-06
>
> We greatly appreciate the reviewer's recognition of our proposed method's effectiveness and broad application scenarios. Regarding the weaknesses raised, we address all the concerns in detail below.
>
> >W1: Writing needs to be calibrated. The sentences in line 64 and line 70 are repeated. 2. The discussion on accelerating convergence is not indexed in the main text (Appendix B) and lacks experimental validation.
>
> We thank the reviewer for pointing out errors in our writing and section arrangement. We will correct the repeated part and discuss more about the convergence in the final manuscript. For the experimental validation, we provide some data points here to prove that FLoRA converges faster than our baselines. We use the Wizard dataset to fine-tune a Llama-7b and test on MT-bench. The LoRA rank here we use is 8. We display the global model performance in the first 5 rounds. FLoRA converges earlier than FedIT (in 3rd round) and has higher accuracy.
>
> =================================
>
> Epochs  $\quad$   0   $\quad$   1   $\quad$  $ $ 2   $\quad$  $ $ 3    $\quad$  $ $ 4   $ $ $\quad$  5
>
> —-------------------------------------------------------
>
> FLoRA  $\ $ $ $ 2.86 $ $ 3.79 $ $ 3.88 $ $ 3.90 $ $ 3.88 $ $ 3.83
>
> FedIT  $\ $ $ $ $ $ 2.86 $ $  2.93 $ $ 3.12 $ $ 3.47 $ $ 3.66 $ $ 3.62
>
> =================================
>
> >W2: More discussion may be needed on the advantages of FLoRA compared to the implementation of 'the aggregation of local update' in Figure 2.
>
> Figure 2 in our paper illustrates the process where the server first recovers the local updates from the LoRA modules and then aggregates them. While this method might be acceptable if the server has ample computational resources, it significantly increases communication costs. Recovering the full gradient on the server necessitates sending back the full model parameter updates to the clients rather than just the LoRA modules. As shown in Figure 6 of our paper, this results in approximately five times more communication cost compared to our FLoRA approach, which is impractical in most scenarios. Therefore, our FLoRA method offers a substantial reduction in communication overhead, making it a more efficient solution than the traditional method of recovering and aggregating local updates on the server.
>
>
> >W3: The foundation models chosen for the experiment are all from the Llama series. Are there other types of foundation models that can be used for validation?
>
> We thank the reviewer for the advice. Due to the limited time of the rebuttal, we are only able to conduct the experiments on homogeneous LoRA fine-tuning on Gemma models. Here are the results of the MT-bench evaluation:
>
> ===========================================
>
> Training Set    $\quad$   $\quad$    $\quad$     Gemma-2b  $\quad$    Gemma-7b
>
> —------------------------------------------------------------------------
>
> MMLU   $\quad$   $\quad$  FLoRA    $\quad$ $\quad$  3.40      $\quad$   $\quad$  $\quad$  5.89
>
> $\quad$  $\quad$ $\quad$  $\quad$ $\ $ FedIT     $\quad$  $ $ $\quad$ 3.33  $\quad$   $\quad$ $\quad$  5.72
>
> Wizard   $\quad$   $\ $ $\ $  FLoRA    $\quad$  $\quad$  3.54   $\quad$   $\quad$ $\quad$  6.65
>
> $\quad$  $\quad$ $\quad$  $\quad$ $\ $ FedIT    $\quad$ $ $ $\quad$  3.44    $\quad$    $\quad$  $\quad$ 6.65
>
> ===========================================
>
> We can see that FLoRA still outperforms the baseline in Gemma models. We will further conduct heterogeneous experiments and experiments of MMLU and ARC evaluation.

---

> > ### Comment · Reviewer_7eRV · 2024-08-08
> >
> > Thanks to the authors for the response and my concerns are well addressed. The simple yet effective method is friendly for practical applications, especially for the federated learning field, so I increase my score.

---

### Official Review · Reviewer_4FRM · 2024-07-13

**Soundness:** 2
**Presentation:** 3
**Contribution:** 3
**Rating:** 5
**Confidence:** 4

**Summary:**

This paper focuses on developing a novel aggregation strategy for training LLMs in FL with LoRA. Specifically, they identify that separately aggregating two matrices of LoRA module is not identical to aggregating model updates. Therefore, they propose a stack based aggregation strategy that merge matrices before aggregating them.

**Strengths:**

1.	Experiments show that the proposed method is effective.
2.	Clear presentation.

**Weaknesses:**

1.	The novelty is limited. Although the proposed method is effective, the contribution of the stack-based aggregation strategy may be too limited.

2.	The motivation of the stack based aggregation is not clear. Since the computational resource of the server is usually sufficient, it is totally acceptable that recovering the local update and then aggregating them.

**Questions:**

no

**Limitations:**

yes

---

> ### Author Rebuttal · Authors · 2024-08-06
>
> We greatly appreciate the reviewer's recognition of the effectiveness of our proposed method. Regarding the weaknesses raised, we address all the concerns in detail below.
>
> >W1: The novelty is limited. Although the proposed method is effective, the contribution of the stack-based aggregation strategy may be too limited.
>
> The primary contribution of our paper is identifying a mathematically inaccurate aggregation noise present in nearly all existing LoRA-based federated fine-tuning algorithms. We then propose a novel and efficient stacking method to address this issue. Our paper introduces significant novelty by being the first to highlight the noise generated by the simplistic averaging process commonly used. Given that almost all current LoRA federated fine-tuning algorithms and codebases rely on this averaging method, our stacking algorithm offers a mathematically accurate and provably convergent solution. We believe this contribution will provide substantial value to the community.
>
>
> >W2: The motivation of the stack based aggregation is not clear. Since the computational resource of the server is usually sufficient, it is totally acceptable that recovering the local update and then aggregating them.
>
> Recovering the local updates and then aggregating them may be acceptable from a computational cost perspective. However, this approach necessitates the server sending back the full model parameter updates to the clients, rather than just the LoRA modules. As illustrated in Figure 6 of our paper, this method results in approximately five times the communication cost compared to our FLoRA approach, which is impractical. Therefore, we argue that recovering the local updates and then aggregating them is not a feasible solution due to the prohibitive communication overhead.

---

> > ### Comment · Reviewer_4FRM · 2024-08-10
> >
> > Thanks for your response. I still have some concerns.
> >
> > 1. I am confused about the necessity of adopting a stack-based data structure to merge the local A_i and B_i modules. It seems that we can just adopt a set to store all modules.
> >
> > 2. It appears that the communication cost is linear to the number of selected clients and the rank of A and B modules. Could the method still save communication costs when the number of selected clients is large or the rank is large?

---

> > > ### Author Response · Authors · 2024-08-10
> > > **Thanks for your response. We have addressed your concerns here.**
> > >
> > > >I am confused about the necessity of adopting a stack-based data structure to merge the local A_i and B_i modules. It seems that we can just adopt a set to store all modules.
> > >
> > > While storing all modules in a set does indeed have the same memory cost as our proposed method, it remains impractical in real-world scenarios for two primary reasons:
> > >
> > > **(1) Incompatibility with LoRA Implementation:** LoRA is not implementable when stored separately. In LoRA-fine-tuned language models, the LoRA modules can be utilized without merging them into the base model parameters, allowing the LoRA modules and the original model parameters to be stored together. As discussed in the LoRA paper [1], the forward pass is represented as:
> > >
> > > $h = \Delta x + W x=W x+BAx,$
> > >
> > > where $BA$ represents the LoRA parameter. In our FLoRA approach, the A and B are global LoRA modules. However, if we store all modules separately, the inference stage would require the computation:
> > >
> > > $h = \Delta x + Wx = Wx + B_0A_0x + B_1A_1x + B_2A_2x + … + B_{K-1}A_{K-1} x,$
> > >
> > > This necessitates applying  $K$ LoRA modules within a single base model, which introduces iterative computations that are inefficient and impractical during the inference stage. Moreover, current LoRA codebases do not support this approach, where a single base model integrates multiple LoRA modules. The inconvenience of using a set to store LoRA modules becomes particularly problematic as the number of clients increases.
> > >
> > > **(2) Incompatibility with Multi-Round Federated Fine-Tuning:** Storing LoRA modules to a set cannot effectively support multi-round federated fine-tuning. In subsequent rounds, fine-tuning requires an updated base model, necessitating updating the base model on the server or the clients. Updating on the server, as discussed in our rebuttal, incurs significantly higher communication costs. Conversely, updating on the clients imposes a substantial computational burden on each client, making this approach impractical.
> > >
> > > >It appears that the communication cost is linear to the number of selected clients and the rank of A and B modules. Could the method still save communication costs when the number of selected clients is large or the rank is large?
> > >
> > > Yes, we are indeed able to save communication costs when using a widely applied LoRA rank.
> > >
> > > LoRA is commonly employed for fine-tuning models because it significantly reduces computational resource demands. This advantage hinges on the premise that the LoRA rank is much smaller than the rank of the model parameters themselves. The key prerequisite for using LoRA is that the number of parameters in LoRA must be substantially smaller than that of the base model. Otherwise, the benefits of LoRA would be negated, making full fine-tuning a more efficient option. Therefore, the LoRA rank should not be so large that it surpasses the communication resources required for transmitting the model’s original parameters.
> > >
> > > Regarding the number of selected clients, it is important to note that the commonly used LoRA rank is typically about 1/100 of the base model size. For instance, in the case of Llama-7b, which has 4096x4096 full parameters, a typical LoRA configuration might use only 16x4096 parameters. This relationship can be expressed as:
> > >
> > > $P_{LoRA} << P_{full}$
> > >
> > > ,where $P$ is the parameter size. When the clients send parameters to the server, FLoRA can consistently save communication costs because:
> > >
> > > $P_{LoRA} < P_{full}$
> > >
> > > This inequality holds true whenever LoRA is used. Moreover, when the server sends parameters back to the clients, FLoRA can also save communication costs under the condition:
> > >
> > > $2K < P_{full}/P_{LoRA}$
> > >
> > > Given that $P_{full}/P_{LoRA}$ is typically greater than 100 in fine-tuning scenarios, and federated fine-tuning server generally lack the communication resources to support extensive client communication (e.g., current works use 10 clients [2]), FLoRA proves effective in reducing communication cost.
> > >
> > > References:
> > >
> > > [1] Hu, E. J., Shen, Y., Wallis, P., Allen-Zhu, Z., Li, Y., Wang, S., ... & Chen, W. (2021). Lora: Low-rank adaptation of large language models. arXiv preprint arXiv:2106.09685.
> > >
> > > [2] Chen, J., Xu, W., Guo, S., Wang, J., Zhang, J., & Wang, H. (2022). Fedtune: A deep dive into efficient federated fine-tuning with pre-trained transformers. arXiv preprint arXiv:2211.08025.

---

> > > > ### Author Response · Authors · 2024-08-12
> > > > **Follow-up on rebuttal**
> > > >
> > > > Dear Reviewer,
> > > >
> > > > As we are approaching the end of the rebuttal period, we would like to cordially inquire about the extent to which we have successfully addressed the concerns outlined in your review.
> > > >
> > > > Should there be any lingering points that require further attention, please rest assured that we are enthusiastic about the opportunity to provide comprehensive responses to any subsequent queries or comments you may have.
> > > >
> > > > Your constructive input remains invaluable to us, and we appreciate your dedication to enhancing the quality of our manuscript. Thank you for your time and consideration.
> > > >
> > > > Best,
> > > > Authors

---

> ### Author Response · Authors · 2024-08-13
> **Looking forward to your response**
>
> Dear Reviewer,
>
> We wanted to inform you that we have addressed the additional concerns you raised. As the rebuttal phase is coming to an end, we kindly request your prompt feedback on our rebuttal.
>
> We appreciate your time and consideration and look forward to your response.
>
> Best regards,
> Authors

---

> > ### Comment · Reviewer_4FRM · 2024-08-13
> >
> > Thanks for the reply. Can the stack-based aggregation save the computational cost?
> > It would be better if the analysis regarding the computational and communication costs could be provided and compared with the aggregation simply using a set.

---

> ### Author Response · Authors · 2024-08-13
> **Thanks for your reply, we clarify the computation and communication cost here**
>
> Compared to storing the LoRA modules in a set, stack-based methods can significantly reduce computational costs due to the following advantages:
>
> 1. **Reduced Memory Overhead**: During the weight recovery process of LoRA, each LoRA module pair is expanded to the full parameter size. In LoRA fine-tuning with \( M \) clients, if the modules are stored in a set, each module must be recovered separately, leading to a total memory usage of \( (M + 1)P_{\text{full}} \), where \( P_{\text{full}} \) represents the full parameter size. This includes the original parameters and \( M \) newly recovered parameters. In contrast, the stack-based method employed in FLoRA allows for the recovery of the full parameters with **only a single matrix multiplication**, resulting in a total memory requirement of just \( 2P_{\text{full}} \). Given that \( M > 1 \) in federated fine-tuning, this approach leads to substantial computational savings.
>
> 2. **Optimized GPU Utilization**: Since parallel computing on a GPU is significantly faster than sequential processing, it is more efficient to compute the stacked LoRA modules collectively rather than individually. This approach better aligns with the GPU’s architecture, leading to faster computations and more efficient resource utilization.
> To illustrate the benefits of our method in terms of computation and communication costs, particularly in comparison to the baseline FedIT and the set-based method proposed by the reviewer, we provide a table below. For reference, the communication and computation costs of FLoRA are normalized to 1:
>
> =========================================================================================
>
> Method $\quad$ Communication  $\quad$ Recovering Computation  $\quad$ Accurate aggregation $\quad$ Applicable to inference
>
> -------------------------------------------------------
>
> FedIT   $\quad$ $\quad$ P_{full}/P_{LoRA}  $\qquad$  $\qquad$ 1  $\qquad$  $\qquad$ $\quad$  $\qquad$  $\qquad$     No $\quad$  $\qquad$  $\qquad$  $\qquad$ $\qquad$ Yes
>
> Set-saving     $\qquad$ $\quad$   1   $\qquad$  $\qquad$ $\qquad$  $(M+1)/2$  $\ $ $\ $ $\ $  $\qquad$ $\quad$ $\qquad$ Yes $\quad$  $\qquad$  $\qquad$  $\qquad$ $\qquad$  No
>
> FLoRA (ours)     $\quad$  $\quad$ 1  $\qquad$ $\qquad$ $\qquad$ $\quad$  1   $\quad$ $\quad$ $\qquad$ $\qquad$ $\quad$ $\qquad$  Yes $\quad$  $\qquad$  $\qquad$  $\qquad$ $\qquad$Yes
>
> =========================================================================================
>
> According to this table, our approach generally outperforms the baselines in important metrics.

---

> ### Author Response · Authors · 2024-08-13
> **Looking forward to your further feedback**
>
> Dear Reviewer 4FRM,
>
> The rebuttal deadline is approaching soon, and we have provided detailed responses to your latest concerns. We kindly request that you review our replies at your earliest convenience. If there are any further questions or issues, please let us know. If our responses have adequately addressed your concerns, we would appreciate a higher score.
>
> Thank you for your time and consideration.

---

> ### Comment · Reviewer_4FRM · 2024-08-14
>
> Thanks for the response.
> I agree that using stack-based aggregation can improve GPU utilization. Yet, I am confused about the other advantages.
>
> 1. The simple set-based aggregation can also have a total memory usage of ( 2P_{\text{full}} ). In particular, we can compute B_iA_i one by one and add outcomes one by one.
>
> 2. I am confused about why the recovering computation of set-based aggregation is (M+1)/2. Considering an extreme case $r=1, A \in R^{d * 1}, B \in R^{1 * d}$, then the computational cost of set-based aggregation is $K*d*d$
>
> The computational cost of stack-based computational cost is also $K*d*d$.

---

> > ### Author Response · Authors · 2024-08-14
> > **Thanks for your response. We further address your concern here.**
> >
> > >The simple set-based aggregation can also have a total memory usage of ( 2P_{\text{full}} ). In particular, we can compute B_iA_i one by one and add outcomes one by one.
> >
> > We do agree that computing B_iA_i one by one has a total memory usage of ( 2P_{\text{full}} ). However, as we mentioned earlier, such sequential computation would significantly increase the time required for the recovery process. If the time for the GPU to perform one $BA$ is t, then calculating for $M$ clients one by one will take approximately $Mt$. This is because the GPUs have strong parallel computing capabilities. The time for the GPU to compute a single LoRA matrix multiplication with $r=1$ is similar to the time required to compute it with $r=M$, whereas performing $M$ separate $r=1$ LoRA matrix multiplications takes $M$ times longer than performing a single $r=M$ multiplication.
> >
> > We also want to recall that the main drawback of using such a set-based aggregation is the difficulty of utilizing LoRA on the base model. The LoRA modules cannot be utilized without merging them into the base model parameters, which limits the implementation of the fine-tuned model.
> >
> > >I am confused about why the recovering computation of set-based aggregation is (M+1)/2. Considering an extreme case $r=1$, then the computational cost of set-based aggregation is $Kdd$. The computational cost of stack-based computational cost is also $Kdd$
> >
> > Here, we need to clarify that the computational overhead we are evaluating primarily concerns GPU memory rather than computational complexity. This is because, in large model fine-tuning/inference, the main bottleneck is memory usage. Current works also mainly focus on reducing memory consumption rather than computational complexity, often even increasing complexity to reduce memory usage (e.g., LoRA). Therefore, our evaluation of computational overhead also focuses on memory consumption. As mentioned in Question 1, if the set-based approach calculates all LoRA modules simultaneously, it will increase memory usage to $(1+M)/2$. However, calculating LoRA modules sequentially would significantly reduce the efficiency of code execution.
> >
> > Thank you for continuing to engage with us, and we look forward to receiving your further feedback.

---

> > > ### Comment · Reviewer_4FRM · 2024-08-14
> > >
> > > Thanks for the response. I agree with the advantage of saving recovery time of the stack-based aggregation. I decide to increase my score to 5. I hope the authors can add this analysis to the revised paper. Besides, the corresponding verification experiments about the recovery efficiency should also be conducted.

---

### Decision · Program_Chairs · 2024-09-25

**Decision:**

Accept (poster)

**Comment:**

This paper proposes a federated learning mechanism to train LLMs with LoRA, The proposal is simple, but practical, with its effectiveness demonstrated in experiments. Therefore, we recommend accepting it.